# Changing trends in traumatic spinal cord injury in an aging society: Epidemiology of 1152 cases over 15 years from a single center in Japan

**Kazuya Yokota**[1,2]*, **Hiroaki Sakai**[1], **Osamu Kawano**[1], **Yuichiro Morishita**[1], **Muneaki Masuda**[1], **Tetsuo Hayashi**[1], **Kensuke Kubota**[1], **Ryosuke Ideta**[3], **Yuto Ariji**[3], **Ryuichiro Koga**[3], **Satoshi Murai**[3], **Ryusei Ifuku**[3], **Masatoshi Uemura**[3], **Junji Kishimoto**[4], **Hiroko Watanabe**[4], **Yasuharu Nakashima**[2], **Takeshi Maeda**[1]

1 Department of Orthopaedic Surgery, Japan Labor Health and Welfare Organization Spinal Injuries Center, Iizuka, Fukuoka, Japan, 2 Department of Orthopaedic Surgery, Graduate School of Medical Sciences, Kyushu University, Higashiku, Fukuoka, Japan, 3 Department of Rehabilitation Medicine, Japan Labor Health and Welfare Organization Spinal Injuries Center, Iizuka, Fukuoka, Japan, 4 Center for Clinical and Translational Research, Graduate School of Medical Sciences, Kyushu University, Higashiku, Fukuoka, Japan

* yokota.kazuya.400@m.kyushu-u.ac.jp

**Data Availability Statement:** The authors cannot share the data publicly due to patient privacy, legal reasons, and it being provided by a third party (in

## Abstract

Traumatic spinal cord injury (TSCI) causes an insult to the central nervous system, often resulting in devastating temporary or permanent neurological impairment and disability, which places a substantial financial burden on the health-care system. This study aimed to clarify the up-to-date epidemiology and demographics of patients with TSCI treated at the largest SCI center in Japan. Data on all patients admitted to the Spinal Injuries Center with TSCI between May 2005 and December 2021 were prospectively collected using a customized, locally designed SCI database named the Japan Single Center Study for Spinal Cord Injury Database (JSSCI-DB). A total of 1152 patients were identified from the database. The study period was divided into the four- or five-year periods of 2005–2009, 2010–2013, 2014–2017, and 2018–2021 to facilitate the observation of general trends over time. Our results revealed a statistically significant increasing trend in age at injury. Since 2014, the average age of injury has increased to exceed 60 years. The most frequent spinal level affected by the injury was high cervical (C1-C4: 45.8%), followed by low cervical (C5-C8: 26.4%). Incomplete tetraplegia was the most common cause or etiology category of TSCI, accounting for 48.4% of cases. As the number of injuries among the elderly has increased, the injury mechanisms have shifted from high-fall trauma and traffic accidents to falls on level surfaces and downstairs. Incomplete tetraplegia in the elderly due to upper cervical TSCI has also increased over time. The percentage of injured patients with an etiology linked to alcohol use ranged from 13.2% (2005–2008) to 19% (2014–2017). Given that Japan has one of the highest aging populations in the world, epidemiological studies in this country will be very helpful in determining health insurance and medical costs and deciding strategies for the prevention and treatment of TSCI in future aging populations worldwide.

this case, Sunfusion Systems). They can provide the relevant URL [http://www.sunfusion.net/index.html]. Owing to the regulations of the Japan Organization of Occupational Health and Safety, there was limited access to the database, and the datasets generated and analyzed during the current study are not publicly available. However, they are available from the corresponding author or the Institutional Review Board of Japan Labor Health and Welfare Organization Spinal Injuries Center (550-4 Igisu, Iizuka, Fukuoka 820-0053, Japan, Tel: +81-948-24-7500, Fax: +81-948-29-1065, E-mail: sic-soumuka01@sekisonh.johas.go.jp) upon reasonable request. Others would be able to access these data in the same manner as the authors. The authors did not have any special access privileges that others would not have.

**Funding:** This work was funded by Japan Society for the Promotion of Science (Grant Number: JP21K16671, awarded to Dr. Kazuya Yokota; Kobayashi Magobe Memorial Medical Foundation, awarded to Dr. Kazuya Yokota; ZENKYOREN (National Mutual Insurance Federation of Agricultural Cooperatives), awarded to Dr. Kazuya Yokota; Ogata Memorial Foundation, awarded to Dr. Kazuya Yokota. The funders had no role in study design, data collection and analysis, decision to publish, or preparation of the manuscript.

**Competing interests:** The authors have declared that no competing interests exist.

**Abbreviations:** ASIA, American Spinal Injury Association; AIS, ASIA Impairment Scale; NLI, neurological level of injury; TSCI, traumatic spinal cord injury; JSSCI-DB, Japan Single Center study for Spinal Cord Injury Database; InSTeP, International Standards Training e-Learning Program; ISNCSCI, International Standard for Neurological Classification of Spina Cord Injury.

## Introduction

Traumatic spinal cord injury (TSCI) is an insult to the central nervous system, often resulting in devastating temporary or permanent neurological impairment and disability [1]. Spinal cord damage results in dysfunction of the motor, sensory, and autonomic systems. Paraplegia is associated with damage to the thoracic, lumbar, or sacral spinal cord, while tetraplegia occurs with damage to the cervical cord. Despite several promising neuroprotective and neurodegenerative strategies in preclinical research, there is currently no highly effective pharmacological or surgical treatment for patients with TSCI [2].

Historically, TSCI has been more common in young adult men and women due to injury sustained in car accidents, falls from a height, and sports [3]. Previous reports on the age distribution of TSCI showed a main peak at 10–29 years over three decades ago; however, the age of the peak group has gradually increased since the 2000s [4, 5]. The reported incidence of TSCI varies from country to country, and even from region to region within the same country [6, 7]. Recent published incidences of TSCI have ranged between 12.6 and 86 per million in the high-income countries [4, 6, 8–13]. In addition, recent SCIs have been characterized by an increase in cervical hyperextension injuries due to falls on level surfaces and low falls in elderly populations [14]. Understanding the up-to-date epidemiology and demographic characteristics of TSCI is crucial for managing future health care resources and identifying prevention guidelines.

Permanent motor, sensory, and autonomic impairments that occur after TSCI have a significant negative impact on the lifestyle of patients and their families. Urinary tract infections, pneumonia, pressure ulcers, and depression resulting from neurological insult can lead to the deterioration of physical and mental health. The more severe the neurological impairment, the longer the rehabilitation required for the SCI patients to return to daily life. Approximately 60%-75% of acute TSCIs involve the cervical spine, 15% the thoracic and 10% the lumbosacral spine [15, 16]. Depending on the level and severity of injury, the costs associated with acute care, rehabilitation, and lost productivity can exceed U.S. $1 million in the first year after injury and up to U.S. $25 million over a lifetime, an annual amount of U.S. $9.7 billion [17, 18]. Understanding the epidemiology and demographics of TSCI is crucial for developing strategies to prevent future TSCIs, as well as for making decisions about post-discharge medical treatment and estimating and/or reducing health care costs.

Reports on the worldwide prevalence and incidence of SCIs indicate significant variations in annual incidence rates, sex ratios, patient age distributions, injury mechanisms, and neurological levels of injury across different countries. Past literature has highlighted Japan as having a notably higher age range among SCI patients compared to other nations [6]. The age-standardized incidence rate of SCI varies based on the socio-demographic Index, reflecting the socioeconomic development level of each country [7]. Developing regions such as South Asia, Saharan Africa, Latin America, the Caribbean, North Africa, and the Middle East experience dramatic linear population growth. In contrast, countries like Japan, with low fertility and mortality rates, exhibit a distinct demographic composition. Consequently, the epidemiology of SCI may differ between low-income and high-income countries. It is highly likely that fertility and mortality rates will decrease in the future in such low-income countries. Therefore, epidemiological studies, especially those in Japan, with unique demographic characteristics, become crucial for understanding global health trends and shaping future healthcare strategies.

According to the World Health Organization (https://www.who.int/), the global population, which increased from 3.6 billion in 1970 to 5.3 billion in 1990, exploded to 8.0 billion by 2022. By 2050, the world's population aged 60 years and older has been estimated to reach over

2.0 billion [19]. The shift in the distribution of a country's population to older age groups began in middle- and high-income countries [19], such as Japan, where more than 30% of its population is over the age of 60 [20]. According to a survey by the Japan Medical Society of Spinal Cord Lesion in 2018, the average age of TSCI patients was 66.5 years, and individuals in their 70s comprised the largest group [12]. The current aging society in Japan reflects future aging populations worldwide [21]. Hence, investigating the epidemiology of TSCI in Japan will be significantly helpful in predicting future global epidemiology of TSCI.

The purpose of this study is to elucidate the current epidemiology and demographics of patients with TSCIs at the largest SCI center in Japan over the past 15 years. Additionally, the study aims to propose future strategies for the prevention of TSCIs.

## Materials and methods

The Spinal Injuries Center in Iizuka City in Fukuoka Prefecture, which opened in 1979, was originally administered by the Japanese Labor Welfare Projects Corporation [22]. The Spinal Injuries Center has a catchment area of approximately 2,000 km$^2$ and serves a population of approximately two million. During the 40-year period since the hospital's opening, our institution has been the only subspecialty SCI center in Japan, and almost all SCIs in Fukuoka Prefecture have been transferred to our institution for assessment and definitive treatment.

Data for all patients with TSCIs admitted to the Spinal Injuries Center between May 2005 and December 2021 were collected using a locally designed SCI database known as the Japan Single Center Study for Spinal Cord Injury Database (JSSCI-DB). Initiated in 2005 at the Spinal Injuries Center, the JSSCI-DB serves as a comprehensive repository. An overview of the JSSCI-DB reveals the collection of longitudinal test data for 134 outcome measures, encompassing neurological assessments, physical function assessments, and health-related quality of life assessments [23, 24]. Various factors, including correct formatting and identification of invalid values, are considered in the assessment of data quality. In the JSSCI-DB, dedicated personnel conduct regular data quality control. The patients' profiles and neurological data used in this study were confirmed after database registration. This study was approved by the Ethical Review Board of Japan Labor Health and Welfare Organization Spinal Injuries Center (Approval code: 16–7). We had all the necessary consent from the patients involved in the study, including consent to participate in the study.

We retrospectively reviewed the dataset for research purposes from July 1 to October 31, 2022. Only patients admitted to our institution within four weeks from injury or fall were included. The admitting surgeons, all of whom were fellowship-trained, dedicated spinal surgeons, and board-certified spinal surgeons approved by the board of the Japanese Society for Spine Surgery and Related Research (http://www.jssr.gr.jp/english/), were responsible for determining the patients' diagnoses, neurology, and treatment strategies. The data were compiled by spinal research coordinators and peer-reviewed by other board-certified spinal surgeons at weekly meetings for accuracy and completeness. Patients with neurological deficits secondary to nontraumatic conditions, such as tumors, infections, vascular abnormalities, cerebrovascular disease, and psychogenic paralysis, who exhibited neurological disorders making it difficult to evaluate the pathology of SCI, were excluded from the study. Patients with medical comorbidities, such as hypertension, diabetes, or heart disease, that did not directly affect severe neurological disorders, were included in this study.

The demographic data collected prospectively included age, sex, height, weight, body mass index (BMI), date of injury, mechanism of injury, length of hospital stay, American Spinal Injury Association (ASIA) Impairment Scale (AIS), spinal or neurological level of injury (NLI), operation rate, and alcohol consumption at injury. The International Standard for the

Neurological Classification of Spinal Cord Injury (ISNCSCI) examinations were conducted by physicians and physician assistants who had completed the ASIA International Standards Training e-Learning Program (InSTeP) as well as in-person training. The mechanism of injury was classified as a fall on a level surface, fall downstairs, traffic accident by car, traffic accident by motorcycle, traffic accident by bicycle, fall from $\geq$ 3m (high fall), fall from < 3 m (low fall), struck by object, sports, and unspecified.

The study period was divided into four- and five-year groups to facilitate the observation of general trends over time depending on the year of the occurrence of the injury: 2005–2009, 2010–2013, 2014–2017, and 2018–2021. Registration in the JSSCI-DB began in the middle of 2005. Between May 2011 and December 2011, the entry of patient data into the database was hindered by hospital renovations. As a result, the number of registered patients considerably decreased in 2011. This study may exhibit variations in the characteristics of the target population due to a temporary suspension of patient admissions in 2011 and a low number of pediatric SCI case admissions, potentially resulting in selection bias. To describe the age groups in this population, we established the following six age ranges: 0–14, 15–29, 30–44, 45–59, 60–74, and more than 75 years [4]. The affected neurological levels were classified as C1-C4, C5-C8, T1-T6, T7-T12, and L1-L5. In addition, the AIS grades of the SCI were categorized as complete or incomplete tetraplegia and complete or incomplete paraplegia. Complete tetraplegia indicates patients with cervical SCIs classified as AIS A, displaying paralysis in both upper limbs and complete paralysis in both lower limbs. Cases of complete paraplegia describe patients with no upper limb paralysis but complete paralysis in both lower limbs. Incomplete tetraplegia includes patients with cervical SCIs encompassing AIS B, C, or D; these individuals exhibit incomplete paralysis in both upper and lower limbs. Incomplete paraplegia refers to patients with no upper limb paralysis but partial paralysis in both lower limbs.

Age, height, weight, and BMI were considered as continuous variables, while age group, sex, mechanism of injury, AIS, NLI, operation rate, alcohol consumption at injury, and category of SCI (tetraplegia and paraplegia) were presented as percentage and treated as categorical variables.

## Statistical analysis

All statistical analyses were performed using the JMP software program version 13 (SAS Institute, Cary, NC, USA). We reported descriptive statistics using proportions for categorical data and means with ranges for continuous data. Additionally, regression analysis was employed for continuous variables such as age, height, weight, and BMI. In contrast, the Cochran-Armitage test was used to analyze trends in qualitative variables, such as proportions. This comprehensive approach enabled us to investigate trends and associations over the 15-year period. We employed a chi-square test to assess the association between alcohol consumption and the NLI in patients with cervical SCI. In all statistical analyses, significance was defined as $P < 0.05$.

## Results

A total of 1152 patients were identified from our database over the 15-year study period from May 2005 to December 2021. A summary of the data on this population is shown in Tables 1 and 2. The absolute number of cases per year ranged from 61 (in 2019) to 88 (in 2014), with a mean of 70 cases, except in 2005 and 2011 (Fig 1A). The percentages of injured patients who were male ranged from 76.4% (in 2021) to 88.5% (in 2008), with a mean of 81.5% (Fig 1B). The mean age of injury was 57.4 years in males and 58.1 years in females, with an age range from 12–92 years. We conducted regression analysis to examine changes

**Table 1. Overall traumatic spinal cord injury demographics (2005–2021, total = 1152).**

| | |
|---|---|
| **Age at injury (year)** | **Male: 57.4 ± 18.7, Female: 58.1 ± 22.0** |
| Sex (n, %) | Male: 935 (81.2), Female: 217 (18.8) |
| Height (cm), Weight (kg), BMI | 164.8 ± 8.65, 61.1 ± 11.9, 22.4 ± 3.54 |
| Level of injury (n, %) | C1-C4: 528 (45.8), C5-C8: 304 (26.4), T1-T6: 36 (3.1), T7-T12: 131 (11.3), L1-L5: 153 (13.3) |
| Mechanism of injury (n, %) | Fall: 717 (62.2), Traffic accident: 282 (24.5), Sports: 57 (4.9), Struck by object: 94 (8.2), Unknown: 2 (0.2) |
| ASIA Impairment Scale at admission (n, %) | A: 307 (26.6), B: 118 (10.2), C: 298 (25.9), D: 258 (22.4), E: 171 (14.8) |
| Tetraplegia (n, %) | Complete: 219 (19), Incomplete: 558 (48.4) |
| Operation rate (n, %) | 601 (52.17) |
| Days to admission after SCI | 3.16 ± 5.14 (0–28), IQR = (0–3) |
| Length of stay (days) | 181.6 ± 146.8 (1–1067), IQR = (55–286) |

ASIA: American Spinal Injury Association; IQR: interquartile range; Variables are given as the mean and standard deviation with the range in parenthesis or as the number with the percentage in parenthesis.

over time in age at TSCI. The results revealed a statistically significant increasing trend in age at injury (Spearman's rank correlation coefficient: age at injury vs. calendar year, estimated regression line: Y = 51.741082 + 0.6322536X, P < 0.0001) (Fig 1C). The age-specific analysis revealed a single peak in TSCI between 60 and 74 years. From 2005–2009, the 45–59 age group was the second most common, but since 2010, the 75+ age group was the second most common group after the 60–74 age group. The percentage of patients 75 years and older increased over time, comprising 26.2% of the patients in the 2018–2021 group (Fig 1D and Table 2). The most frequent spinal level affected by the injury was high cervical (C1-C4: 45.8%), followed by low cervical (C5-C8: 26.4%), lumbar (L1-L5: 13.3%), high thoracic (T1-T6: 3.1%), and low thoracic (T7-T12: 11.3%). Throughout the 2005–2021 period, cervical TSCI accounted for 70% or more of all TSCI patients (Fig 1E, Tables 1 and 3).

Variables are given as the mean and standard deviation with the range in parenthesis or as the number with the percentage in parenthesis.

**Table 2. Demographic profile of patients with TSCI.**

| | 2005–2009 | 2010–2013 | 2014–2017 | 2018–2021 | |
|---|---|---|---|---|---|
| Sample size (n) | 334 | 250 | 305 | 263 | P value |
| Age at injury | 52.6 ± 20.3 | 58.4 ± 19.6 | 60.2 ± 17.7 | 60. 0 ± 18.7 | < 0.0001 |
| **Age group** | | | | | |
| 0–14 | 2 (0.6) | 1 (0.4) | 0 (0) | 1 (0.4) | 0.4452 |
| 15–29 | 60 (18) | 32 (12.8) | 26 (8.5) | 24 (9.1) | 0.0002 |
| 30–44 | 45 (13.5) | 30 (12) | 28 (9.2) | 30 (11.4) | 0.2416 |
| 45–59 | 75 (22.5) | 29 (11.6) | 49 (16.1) | 43 (16.3) | 0.089 |
| 60–74 | 96 (28.7) | 102 (40.8) | 125 (41) | 96 (36.5) | 0.0295 |
| 75+ | 56 (16.8) | 56 (22.4) | 77 (25.2) | 69 (26.2) | 0.0029 |
| Male/Female | 278 (83.2)/56 | 203 (81.2)/47 | 241 (79)/64 | 213 (81)/50 | 0.3385 |
| Height (cm) | 164.6 ± 8.71 | 164.2 ± 8.1 | 164.8 ± 8.9 | 165.4 ± 8.78 | 0.2309 |
| Weight (kg) | 61.4 ± 11.3 | 60.7 ± 12.2 | 59.9 ± 11.5 | 62.7 ± 12.7 | 0.4708 |
| BMI (kg/m$^2$) | 22.6 ± 3.25 | 22.4 ± 3.67 | 22 ± 3.51 | 22.8 ± 3.76 | 0.9174 |
| Operation rate | 183 (54.8) | 116 (46.4) | 145 (47.5) | 157 (59.7) | 0.3839 |

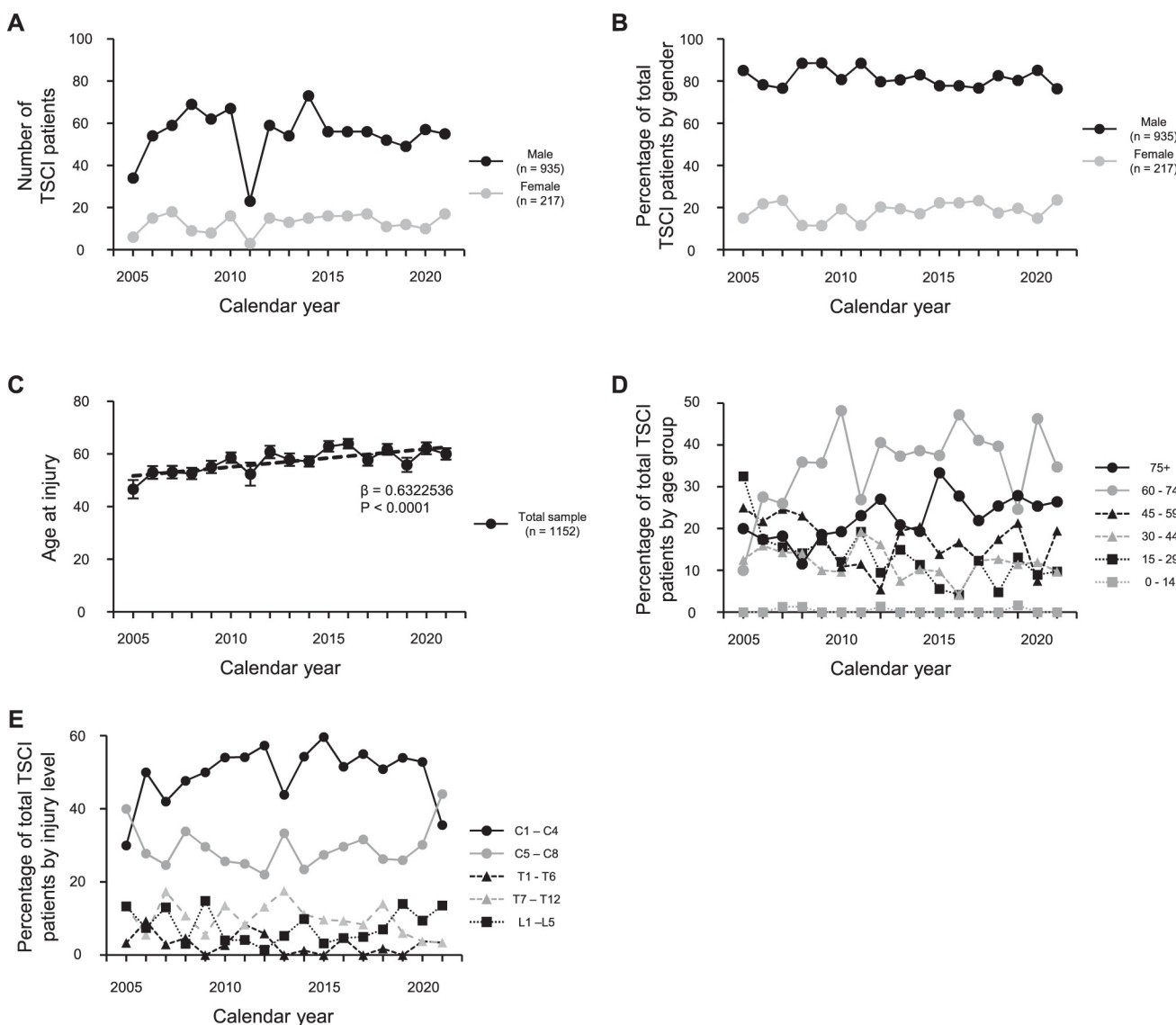

**Fig 1.** (A and B) Number (A) and percentage (B) of traumatic spinal cord injuries (TSCI) from 2005 to 2021 (n = 1152). The incidence rate was consistently higher in men than in women. Registration at the Japan Single Center Study for Spinal Cord Injury Database (JSSCI-DB) began in the middle of 2005. Patient data could not be entered into the database because of hospital renovations from May 2011 to December 2011; therefore, the number of registered patients decreased considerably in 2005 and 2011. (C) Temporal change in average age at injury from 2005 to 2021. Over time, there is a significant increasing trend in age at injury (Spearman correlation coefficient, β = 0.6322536, P < 0.0001). (D) Percentages of TSCI by age group from 2005 to 2021. The 60–74 age group was the most common group, and the 75+ age group was the second most common, beginning in 2010. (E) Percentages of TSCI by neurological level of injury (NLI) from 2005 to 2021. The most frequent NLI affected by the injury was the high cervical level (C1-C4), followed by the low cervical level (C5-C8).

Regarding the extent of injury, 26.6% of the cases were classified as AIS grade A, 10.2% as grade B, 25.9% as grade C, 22.4% as grade D, and 14.8% as grade E. AIS grade A was the most prevalent in all TSCIs. Notably, the percentage of grade D TSCIs increased dramatically from 2005–2009 (7.8%) to 2018–2021 (38.8%) (P < 0.0001). Incomplete tetraplegia was the most common category of SCI, accounting for 48.4% of the cases (Table 1). The percentage of SCIs as incomplete tetraplegia also increased progressively from 2005–2009 (43.7%) to 2018–2021 (52.1%) (P = 0.0295), while the percentage of SCIs as complete paraplegia decreased from 2005–2009 (11.7%) to 2018–2021 (4.2%) (P = 0.0002) (Table 3).

**Table 3. Injury profile of patients with TSCI.**

| Characteristics | Injury Year Intervals | | | | |
| --- | --- | --- | --- | --- | --- |
| | **2005–2009** | **2010–2013** | **2014–2017** | **2018–2021** | **P value** |
| Sample size (n) | 334 | 250 | 305 | 263 | NA |
| Etiology of injury | | | | | |
| Fall on level surface (n, %) | 44 (13.2) | 37 (14.8) | 68 (22.3) | 49 (18.6) | 0.0116 |
| Fall downstairs (n, %) | 18 (5.1) | 22 (8.8) | 26 (8.5) | 33 (12.5) | 0.0037 |
| Traffic accident car (n, %) | 39 (11.7) | 49 (19.6) | 34 (11.1) | 28 (10.6) | 0.2889 |
| Traffic accident motorcycle (n, %) | 28 (8.4) | 9 (3.6) | 24 (7.9) | 12 (4.6) | 0.2195 |
| Traffic accident bicycle (n, %) | 16 (4.8) | 16 (6.4) | 9 (3) | 18 (6.8) | 0.6755 |
| High fall (n, %) | 56 (16.8) | 43 (17.2) | 42 (13.8) | 43 (16.3) | 0.6 |
| Low fall (n, %) | 79 (23.7) | 40 (16) | 71 (23.3) | 46 (17.5) | 0.2507 |
| Struck by object (n, %) | 30 (9) | 21 (8.4) | 21 (6.9) | 22 (8.4) | 0.5975 |
| Sports (n, %) | 24 (7.2) | 13 (5.2) | 9 (3) | 11 (4.2) | 0.0348 |
| Unspecified or unknown (n, %) | 0 (0) | 0 (0) | 1 (0.3) | 1 (0.4) | NA |
| Alcohol consumption at injury | | | | | |
| Yes | 44 (13.2) | 41 (16.4) | 58 (19) | 42 (16) | 0.1974 |
| No | 290 (86.8) | 207 (82.8) | 240 (78.7) | 214 (81.4) | |
| Unspecified or unknown | 0 (0) | 2 (0.8) | 7 (2.3) | 7 (2.7) | NA |
| Neurological level of injury | | | | | |
| C1–C4 | 123 (45.2) | 117 (52.5) | 147 (55.1) | 105 (47.9) | 0.268 |
| C5–C8 | 82 (30.1) | 59 (26.5) | 74 (27.7) | 70 (32.0) | 0.5609 |
| T1–T6 | 11 (4) | 8 (3.6) | 4 (1.5) | 5 (2.3) | 0.154 |
| T7–T12 | 29 (10.7) | 31 (13.9) | 26 (9.7) | 15 (6.8) | 0.6511 |
| L1–L5 | 27 (9.9) | 8 (3.6) | 16 (6) | 24 (11) | 0.2196 |
| AIS/Frankel grade | | | | | |
| A | 98 (29.3) | 77 (30.8) | 83 (27.2) | 49 (18.6) | 0.0036 |
| B | 36 (10.8) | 28 (11.2) | 29 (9.5) | 25 (9.5) | 0.4973 |
| C | 112 (33.5) | 67 (26.8) | 76 (24.9) | 43 (16.3) | < 0.0001 |
| D | 26 (7.8) | 51 (20.4) | 79 (25.9) | 102 (38.8) | <0.0001 |
| E | 62 (18.6) | 27 (10.8) | 38 (12.5) | 44 (16.7) | 0.4744 |
| Category of SCI | | | | | |
| Complete tetraplegia | 59 (17.7) | 57 (22.8) | 65 (21.3) | 38 (14.4) | 0.3141 |
| Incomplete tetraplegia | 146 (43.7) | 119 (47.6) | 156 (51.1) | 137 (52.1) | 0.0295 |
| Complete paraplegia | 39 (11.7) | 20 (8) | 18 (5.9) | 11 (4.2) | 0.0002 |
| Incomplete paraplegia | 28 (8.4) | 27 (10.8) | 28 (9.2) | 33 (12.5) | 0.1902 |

NA: not applicable; Variables are given as the mean with the percentage in parenthesis.

The operation rate of cases was 52.17% throughout the study period, with the highest percentage of operation rate in 2018–2021 (59.7%) (Tables 1 and 2). The average length of stay in our institution was 181.6 (s.d. ± 146.8) days, and the average time to admission after SCI was 3.16 (s.d. ± 5.14) days (Table 1).

The most frequent causes of TSCI during the study period were falls (62.2%), followed by traffic accidents (24.5%), struck by objects (8.2%), and sports (4.9%) (Table 1). The analysis of injury mechanisms revealed that falls on a level surface were the most common in 2018–2021, while high or low falls were the most common injury mechanism from 2005–2009 (Table 3). Regarding the characteristics of the etiological distribution of the injuries over the study period, the most relevant changes were an increase in the number of falls on level surfaces

(P = 0.0116) and downstairs (P = 0.0037), and a decrease in the number of falls during sports (P = 0.0348) (Table 3). The percentage of injured patients with an etiology linked to alcohol use ranged from 13.2% (2005–2008) to 19% (2014–2017), with a mean of 16.1% (Table 3). In addition, we analyzed trends in alcohol consumption, neurological level of injury (NLI), and severity of injury for each age group. No discernible increasing or decreasing trend over time was observed regarding the percentage of those who had consumed alcohol at the time of injury (P = 0.1974). In terms of NLI, C1-C4 and C5-C8 cervical TSCIs accounted for the majority of cases, with no observable trend of increase or decrease over time (C1-C4: P = 0.268; C5-C8: P = 0.5609). In terms of the severity of SCI, there was a decreasing trend over time for AIS grade A (P = 0.0036) and grade C (P < 0.0001), while an increasing trend was observed for grade D (P < 0.0001).

We observed a significant increase in the likelihood of patients experiencing falls on level surfaces (P < 0.0001), descending stairs (P < 0.0001), or low-level falls (P < 0.0001) as the age of injury increased. Conversely, a younger age of injury was significantly associated with traffic accidents involving motorcycles (P < 0.0001), falls from heights (P < 0.0001), being struck by objects (P = 0.0036), or sports injuries (P < 0.0001). The older the patient was at the time of injury, the more likely they were to consume alcohol (P = 0.0052). Additionally, with increasing age of injury, older patients tended to have injuries at the C1-C4 cervical level (P < 0.0001), while younger patients exhibited a tendency for more thoracic (T1-T6: P = 0.0003; T7-T12: P < 0.0001) and lumbar (L1-L5: P < 0.0001) spine injuries. As the age at the time of injury increased, there was a higher percentage of AIS grade C (P = 0.0132) and D (P = 0.0002) cases, while grade A (P = 0.0006) cases tended to be more common in younger patients (Table 4). We further explored the association between neurological level of injury (NLI) and alcohol consumption. When restricting our analysis to patients with cervical cord injuries and NLI at C1-C4 and C5-C8, we observed a significant correlation between alcohol consumption and increased damage in the upper cervical cord (odds ratio = 1.64, 95% confidence interval 1.13–2.37, p = 0.0084) (S1 Table).

The peak days for the occurrence of TSCIs were Friday, Saturday, and Sunday. Throughout the entire 2005–2021 period, no statistically significant trend was observed in terms of the day of injury (Fig 2A and S2 and S3 Tables). Seasonal variations in TSCI incidences differed from year to year, and TSCIs occurred more often during the fall months and least frequently in the winter months. According to the statistical results, there was a trend over the 15-year period indicating an increase in the percentage of injured patients in January (P = 0.0481) and October (P = 0.014). TSCIs were the most common in October and the least common in February (Fig 2B and 2C, S4 and S5 Tables).

## Discussion

In this study, we conducted a detailed investigation of the epidemiology and demographics of TSCI over a period of 15 years at a single institution in Japan. The mean age at injury increased over time, demonstrating an increasing trend in the proportion of elderly TSCIs over 60 years. A characteristic feature of TSCI in the elderly was that the injury was at the cervical level (C1-C4 or C5-C8), and the severity of injury included a comparatively large portion of TSCIs with mild neurological impairment, such as AIS C or D.

The mean age at injury in the most recent period from 2018–2021 was 60 years, whereas the mean age was 66.5 years in a Japanese nationwide epidemiological study conducted in 2018 [12]. Furthermore, the percentage of falls on level surfaces accounted for 18.6% in this study, corresponding to about one-half of the number (38.6%) in the nationwide study. Because our facility is actively involved in the social reintegration process after SCI in younger

**Table 4. Injury profile of patients with TSCI in each age group.**

| Characteristics | Age group | | | | | | |
|---|---|---|---|---|---|---|---|
| | 0–14 | 15–29 | 30–44 | 45–59 | 60–74 | 75+ | P value |
| Etiology of injury | | | | | | | |
| Fall on level surface (n, %) | 0 (0) | 1 (0.5) | 7 (3.5) | 28 (14.1) | 84 (42.4) | 78 (39.4) | < 0.0001 |
| Fall downstairs (n, %) | 0 (0) | 1 (1.0) | 2 (2.0) | 12 (12.1) | 50 (50.5) | 34 (34.3) | < 0.0001 |
| Traffic accident car (n, %) | 0 (0) | 28 (18.7) | 16 (10.7) | 28 (18.7) | 38 (25.3) | 40 (26.7) | 0.1826 |
| Traffic accident motorcycle (n, %) | 0 (0) | 21 (28.8) | 17 (23.3) | 17 (23.3) | 15 (20.5) | 3 (4.1) | < 0.0001 |
| Traffic accident bicycle (n, %) | 0 (0) | 3 (5.1) | 5 (8.5) | 12 (20.3) | 28 (47.5) | 11 (18.6) | 0.1806 |
| High fall (n, %) | 1 (0.5) | 35 (19.0) | 31 (16.8) | 36 (19.6) | 64 (34.8) | 17 (9.2) | < 0.0001 |
| Low fall (n, %) | 0 (0) | 10 (4.2) | 17 (7.2) | 35 (14.8) | 112 (47.5) | 62 (26.3) | < 0.0001 |
| Struck by object (n, %) | 0 (0) | 7 (7.4) | 27 (28.7) | 24 (25.5) | 25 (26.6) | 11 (11.7) | 0.0036 |
| Sports (n, %) | 3 (5.3) | 35 (61.4) | 10 (17.5) | 4 (7.0) | 3 (5.3) | 2 (3.5) | < 0.0001 |
| Unspecified or unknown (n, %) | 0 (0) | 1 (50) | 1 (50) | 0 (0) | 0 (0) | 0 (0) | NA |
| Alcohol consumption at injury | | | | | | | |
| Yes (n, %) | 0 (0) | 7 (3.8) | 17 (9.2) | 36 (19.5) | 97 (52.4) | 28 (15.1) | 0.0052 |
| No (n, %) | 4 (0.4) | 135 (14.2) | 115 (12.1) | 158 (16.6) | 312 (32.8) | 227 (23.9) | |
| Unspecified or unknown (n, %) | 0 (0) | 0 (0) | 1 (6.3) | 2 (12.5) | 10 (62.5) | 3 (18.8) | NA |
| Neurological level of injury | | | | | | | |
| C1-C4 (n, %) | 2 (0.4) | 22 (4.2) | 29 (5.5) | 86 (16.3) | 242 (45.8) | 147 (27.8) | < 0.0001 |
| C5-C8 (n, %) | 1 (3.3) | 50 (16.4) | 33 (10.9) | 43 (14.1) | 103 (33.9) | 74 (24.3) | 0.3431 |
| T1-T6 (n, %) | 0 (0) | 10 (27.8) | 7 (19.4) | 9 (25.0) | 5 (13.9) | 5 (13.9) | 0.0003 |
| T7-T12 (n, %) | 1 (0.8) | 23 (17.6) | 27 (20.6) | 26 (19.8) | 34 (26.0) | 20 (15.3) | < 0.0001 |
| L1-L5 (n, %) | 0 (0) | 37 (24.2) | 37 (24.2) | 32 (20.9) | 35 (22.9) | 12 (7.8) | < 0.0001 |
| AIS/Frankel grade | | | | | | | |
| A | 2 (0.7) | 53 (17.3) | 39 (12.7) | 51 (16.6) | 105 (34.2) | 57 (18.6) | 0.0006 |
| B | 1 (0.8) | 17 (14.4) | 10 (8.5) | 16 (13.6) | 44 (37.2) | 30 (25.4) | 0.7005 |
| C | 0 (0) | 23 (7.7) | 33 (11.1) | 53 (17.8) | 120 (40.3) | 69 (23.2) | 0.0132 |
| D | 0 (0) | 22 (8.5) | 17 (6.6) | 45 (17.4) | 105 (40.7) | 69 (26.7) | 0.0002 |

NA: not applicable; Variables are given as the mean with the percentage in parenthesis.

patients [25], the proportion of patients with TSCI due to traffic trauma and sports injuries remained high, leading to a decrease in the proportion of elderly patients with TSCI in our database. Cervical SCI in the elderly increases the likelihood of developing complications, such as pneumonia, urinary tract infection, and decubitus ulcers [26], which may imply that elderly cases could not receive satisfactory treatment in specialized facilities such as ours because of the priority given to the systemic management of the treatment of such complications. This study is one of the few to accurately show the percentage of TSCIs in patients who were consuming alcohol at the time of injury [27]. The findings showed that 13.2% to 19% of the patients had consumed alcohol. Patients with cervical SCI who consumed alcohol at the time of injury had a 1.64 times greater risk of upper cervical SCI with NLI of C1-C4. Therefore, avoiding excessive alcohol consumption may help prevent SCIs in the elderly. Additionally, reducing the risk of upper cervical SCI could potentially lead to a reduction in healthcare costs. Because populations worldwide are aging, the recent increase in cervical SCI in the elderly will be an important issue that could accelerate the pressure on health care costs in many countries, including currently low- and middle-income countries [28].

This epidemiologic study was based on prospectively collected data on more than 1000 patients with acute TSCI who were admitted to our institution, which is the largest reported

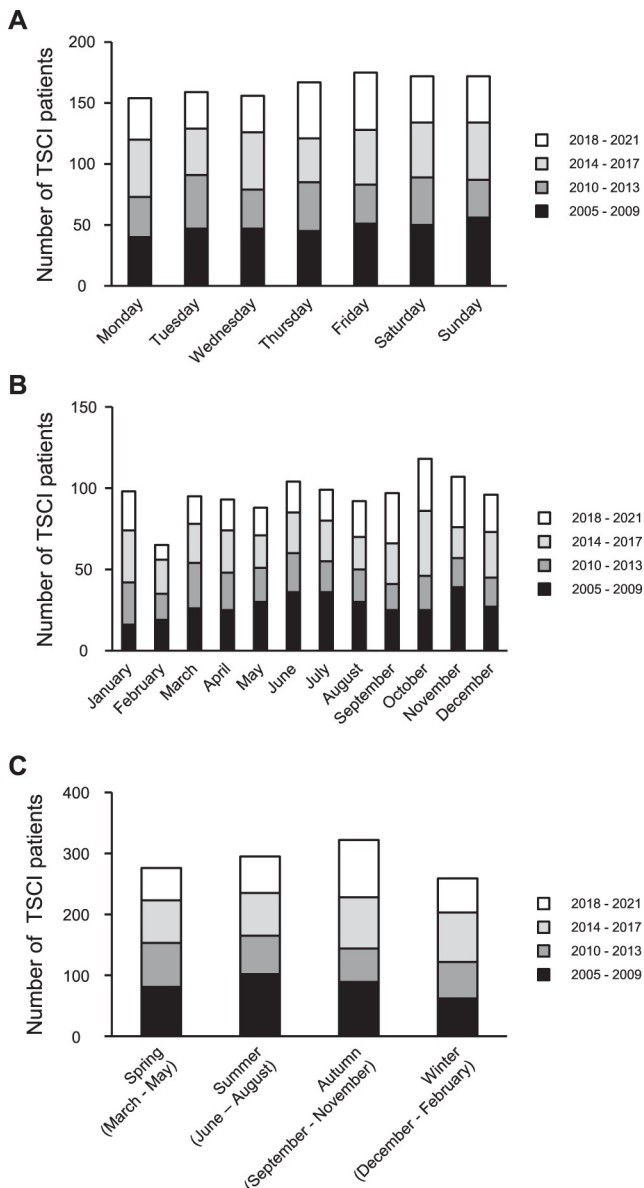

**Fig 2.** (A) Weekday variations in injuries and the number of TSCI cases per weekday (n = 1152). The peak days for the occurrence of TSCI were Friday, Saturday, and Sunday. Throughout the entire period, there were more than 150 cases each day of the week. (B and C) Seasonal variations in injuries and the number of TSCI cases per month. TSCIs were most common in October and least common in February. TSCIs occurred more often during the fall months and were the least common in the winter months.

population-based sample of TSCI in single institution in Japan. The single-institution database has advantages over multicenter studies. First, all examiners who collected neurological findings in this study had completed formal InSTeP training, and the decided AIS grades and NLIs were subsequently reviewed by similarly qualified colleagues, which strongly ensured the quality of this database. In collecting neurological data on TSCI in a multicenter study, because previous assessors may not have completed formal examination training, they would have been less reliable in determining neurological outcomes. Indeed, Kirshblum and colleagues presented challenging classifications along with responses and explanations [29, 30], even

though the ISNCSCI endorsed by ASIA and the International Spinal Cord Society (ISCoS) has undergone revision and updates over time [31]. For instance, in some challenging cases, patients have been diagnosed with AIS A due to the lack of sacral sparing, although from the standpoint of walking ability, the patients appear to have AIS D-like capability because muscle strength was significantly preserved [32]. Second, we prospectively entered the neurological findings of all patients in electronic medical records and determined both NLI and AIS grades according to the exact diagnostic criteria officially sanctioned by ASIA. According to Miya-koshi and colleagues, the nationwide survey was based on a retrospective design, which may have resulted in missing values and an insufficient amount of data. Integrating our database system into core spinal centers across the country will lead to increasing the number and accuracy of nationwide epidemiological studies of TSCI. A previous report suggested inaccuracies in the manual ISNCSCI worksheets in the clinical setting and 75% of ISNCSCI worksheets had one or more errors when completed manually while undertaking randomized controlled trials [33]. Osunronbi and colleagues indicated that the quality of ISNSCI documentation remained poor, and they proposed that reasons for the low quality of ISNSCI documentation included negligence and lack of knowledge [34]. Well-established training and a computerized algorithm may be necessary to ensure accurate scoring, scaling, and classification of the ISNSCI. Third, our facility is unique because after incorporating them into our database from the acute stage of SCI, we were able to follow up patients in the same institution over a long period. The average length of hospital stay was more than 180 days. The average time of admission of the patients shown in this study was 3.16 days (0–28 days) after injury. In contrast, most previous demographic studies did not specify the time to hospitalization after injury. We believe that our facility will play a critical role in future clinical trials in Japan, as it can provide long-term follow-up data on acute operative and non-operative care, as well as chronic rehabilitation care following SCI.

According to reports on the global prevalence and incidence of SCI, the peak age of SCI patients is in the 20s to 40s, with traffic trauma being the most common etiology of injury in Spain, Turkey, and Greece [6]. In a single-center study in China, the age of injury peaked at 40 years, although there was an increasing trend in the age of injury from year to year [35]. The mean age of injury in Scotland was approximately 50 years [36]. There is concern that the age of injury may increase in the future in the above-mentioned low-middle-income countries. In fact, the results of this study show that the distribution of the injured population is shifting to a single peak of 60 to 74 years of age. In Germany, a leading country in terms of aging, the most recent report indicates that the peak age of injury is changing to 75–80 years old [37]. Injury ages and patterns are likely to differ depending on the population distribution and living environment in each country, and the results of this study are expected to reflect the future outlook for other low-middle-income countries.

According to the results of this study, there is a noteworthy increasing trend in the age at injury among TSCI patients over time. Additionally, there is a significant rise in the incidence of certain mechanisms of injury, particularly falls on level ground, falls downstairs, and falls from low heights. Conversely, there is a decline in sports-related injuries among young individuals, which previously constituted a substantial proportion of TSCI cases. With the ongoing global progression toward an aging society, it is highly anticipated that the number of SCI patients in the elderly demographic will rise significantly [7]. Consequently, it is imperative to develop future treatment strategies for elderly SCI patients, taking into account aspects such as surgical interventions, rehabilitation, and preventive medicine.

A global survey of TSCI has highlighted significant differences in the mechanisms of injury between high-income and low- to middle-income countries [6]. Our findings and previous reports showed that the majority of TSCI result from falls on level ground in Japan [11, 12],

while low- to middle-income countries experience a particularly high frequency of TSCI caused by violence. An epidemiological study in Colombia revealed that interpersonal violence is the most common mechanism of TSCI in the region [38]. Notably, the severity of TSCI resulting from interpersonal violence often leads to a high percentage of ASIA grade A (complete paralysis). In low- to middle-income countries, poor outcomes are exacerbated by factors such as prolonged transport times for injured patients, limited access to intensive care units for treatment, and a high proportion of treatment costs [39, 40]. Strikingly, 90% of these injuries in Colombia are attributed to gun violence. In response, Colombia is implementing a municipal gun control program with the hope of developing more effective methods in the future. In Japan, successful prevention strategies involve awareness-raising activities for fall prevention and improvements in the living environment. However, in low- to middle-income countries with diverse causes of TSCI, the development of entirely different prevention strategies is imperative.

This study has several limitations. First, because the registration of the database began in the middle of 2005, and the hospital was renovated in 2011, the data for these years are not complete. Second, the patients' hospitalization periods ranged from 0–28 days after their injury. Within four weeks of an injury, the AIS grade could change significantly, and even small differences in the time of evaluation could affect the neurological findings on admission. Third, our facility did not deal with pediatric trauma cases under 14 years of age. The very small number of pediatric TSCI cases may have resulted in an increase in the average age at the time of injury. This study is a single-center analysis and hence may have limited generalizability. Finally, the data in this study included information under patent, and not all of the data could be made publicly available. Despite these limitations, this study represents large single-center data from Japan, a representative of an aging society, and will help develop future treatment strategies for SCI patients in an aging society worldwide. The age at SCI is anticipated to rise globally in the future, emphasizing the need to establish preventive measures and treatment strategies for SCIs in the elderly. Additionally, exploring the potential impact of interventions aimed at controlling excessive alcohol consumption and preventing falls on reducing the incidence of SCI is desirable.

## Conclusion

To accurately determine the epidemiology of SCI patients, it is necessary to establish an environment that allows for the long-term, consistent follow-up of patients in specialized facilities by professional staff. This study presents the prospective findings of the demographics and epidemiology of TSCI in single center in Japan over a period of more than 15 years since 2005. Throughout the study period, approximately 80% of the patients were male. Since 2014, the average age at injury has gradually increased to exceed 60 years. As the number of injuries among the elderly increased, the injury mechanisms shifted from high-fall trauma and traffic accidents to falls on level surfaces and downstairs. Incomplete tetraplegia in the elderly due to upper cervical SCI also increased over time. Avoiding excessive alcohol consumption may help prevent SCIs in the elderly. Given that Japan has one of the highest aging populations in the world, epidemiological studies in this country will be very helpful in determining health insurance and medical costs and deciding treatment strategies for SCI in future aging populations worldwide.

## Supporting information

**S1 Checklist. Human participants research checklist.**
(DOCX)

**S1 Table. The relationship between the level of injury and alcohol consumption at the time of injury.**
(DOCX)

**S2 Table. Weekday variations in injuries and the number of TSCI cases per month based on the date of injury.**
(DOCX)

**S3 Table. Weekday variation in injuries and the percentage of TSCI cases per month based on the date of injury.**
(DOCX)

**S4 Table. Seasonal variations in injuries and the number of TSCI cases per month based on the date of injury.**
(DOCX)

**S5 Table. Seasonal variations in injuries and the percentage of TSCI cases per month based on the date of injury.**
(DOCX)

## Author Contributions

**Conceptualization:** Kazuya Yokota.

**Data curation:** Kazuya Yokota, Hiroaki Sakai, Osamu Kawano, Yuichiro Morishita, Muneaki Masuda, Tetsuo Hayashi, Kensuke Kubota, Ryosuke Ideta, Yuto Ariji, Ryuichiro Koga, Satoshi Murai, Ryusei Ifuku, Masatoshi Uemura.

**Formal analysis:** Kazuya Yokota, Hiroaki Sakai, Osamu Kawano, Yuichiro Morishita, Muneaki Masuda, Tetsuo Hayashi, Kensuke Kubota, Ryosuke Ideta, Yuto Ariji, Ryuichiro Koga, Satoshi Murai, Ryusei Ifuku, Masatoshi Uemura, Junji Kishimoto, Hiroko Watanabe.

**Funding acquisition:** Kazuya Yokota.

**Investigation:** Kazuya Yokota, Hiroaki Sakai, Osamu Kawano, Yuichiro Morishita, Muneaki Masuda, Tetsuo Hayashi, Kensuke Kubota, Ryosuke Ideta, Yuto Ariji, Ryuichiro Koga, Satoshi Murai, Ryusei Ifuku, Masatoshi Uemura, Junji Kishimoto, Hiroko Watanabe.

**Methodology:** Kazuya Yokota, Junji Kishimoto, Hiroko Watanabe.

**Project administration:** Kazuya Yokota, Hiroaki Sakai, Yasuharu Nakashima, Takeshi Maeda.

**Resources:** Kazuya Yokota, Yasuharu Nakashima, Takeshi Maeda.

**Software:** Kazuya Yokota.

**Supervision:** Kazuya Yokota, Yasuharu Nakashima, Takeshi Maeda.

**Validation:** Kazuya Yokota.

**Visualization:** Kazuya Yokota.

**Writing – original draft:** Kazuya Yokota.

**Writing – review & editing:** Kazuya Yokota.

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
