## [Decision Letter · Decision Letter 0]

31 Oct 2023

PONE-D-23-30026Changing trends in traumatic spinal cord injury in an aging society: epidemiology of 1152 cases over 15 years from a single center in JapanPLOS ONE

Dear Dr. Yokota,

Thank you for submitting your manuscript to PLOS ONE. After careful consideration, we feel that it has merit but does not fully meet PLOS ONE’s publication criteria as it currently stands. Therefore, we invite you to submit a revised version of the manuscript that addresses the points raised during the review process.

 Both reviewers see the impact of this study. However, they raised several major issues regarding statistical analysis. Please respond to these comments and make needed changes to data analyses. 

We look forward to receiving your revised manuscript.

Kind regards,

Yih-Kuen Jan, PhD

Academic Editor

PLOS ONE

Reviewers' comments:

Reviewer's Responses to Questions

**Comments to the Author**

1. Is the manuscript technically sound, and do the data support the conclusions?

Reviewer #1: Partly

Reviewer #2: No

2. Has the statistical analysis been performed appropriately and rigorously? 

Reviewer #1: I Don't Know

Reviewer #2: No

3. Have the authors made all data underlying the findings in their manuscript fully available?

Reviewer #1: Yes

Reviewer #2: Yes

4. Is the manuscript presented in an intelligible fashion and written in standard English?

Reviewer #1: Yes

Reviewer #2: Yes

5. Review Comments to the Author

Reviewer #1: The authors performed a study addressing “Changing trends in traumatic spinal cord injury in an aging society: epidemiology of 1152 cases over 15 years from a single center in Japan”. However, there remain several concerns to be clarified, some of which are critical.

1. It is unclear what you want to see in this study. “The purpose of this study is to clarify changes in the demographic and epidemiologic characteristics of individuals with TSCI over the past 15 years based on age in a single institute in Japan. changes in the demographic and epidemiologic characteristics of individuals with TSCI over the past 15 years at a single institute in Japan based on age distribution, sex, cause of injury, level of injury, severity of injury, and seasonality in the number of patients.” (p.6 line 6-10) The purpose is not clear in the results of this study. Please correct it.

2. Please describe the results (including tables and figures), including the site of injury and cause of injury by age group.

3. Gradually getting older, is there a significant difference? It would seem that a statistical difference should be sought.

4. The χ2 is mentioned in the analysis method, but it is unclear whether it is actually done or not.

5. I do not see where there is a significant difference in Table 1, 2, and 3. Please describe the tables more clearly.

Reviewer #2: This scientific article, titled "Epidemiology and Demographics of Traumatic Spinal Cord Injury in Japan," provides valuable insights into the epidemiology and demographics of traumatic spinal cord injuries (TSCI) in Japan. This article presents a comprehensive epidemiological study of traumatic spinal cord injuries (TSCI) over a 15-year period, focusing on patients treated at the Spinal Injuries Center in Japan. The study reveals a shift in the demographics of TSCI, with a notable increase in elderly patients, highlighting the aging population in Japan. The most frequent spinal level affected was the cervical region, and the severity of injuries ranged from complete to incomplete tetraplegia. Falls emerged as the leading cause of TSCI, with a substantial portion of injuries related to alcohol consumption. The study identifies rising health care challenges and emphasizes the importance of understanding the evolving epidemiology of TSCI for resource management and prevention strategies, especially in the context of aging populations worldwide.

The study provides a comprehensive and detailed analysis of traumatic spinal cord injuries (TSCI) over a 15-year period, making use of data from more than 1,000 patients. The use of a single-institution database ensures consistency and accuracy in data collection and analysis as well as the longitudinal data provides a bigger picture of the issue of TSCI.

However, there are some notable flaws in the article that I would like to describe below:

Major comments:

•There is an evident information bias in the periods where the data was not collected properly. For accurate data analysis you should remove that data from this article. In that same line of ideas, the statistical analysis is not strong enough for the aim and the data use that is described. Using chi2 limits the study comparability and conclusions.

• in the methods section, chi square tests are mentioned. However, there is no p values reported in the results section of this study. Was an analysis carried out?

• in the results section, is it a percentage of the total patients presenting to the facility? Or is it the total patients with TSCI? Please clarify to avoid confusion. The axis titles can be used to bring this out well.

• Would it be possible to carry out regression analysis in order to get odds ratios? These may prove to be beneficial to thee analyses and results section. These would also help to account for effect modifiers like operation rates on length of stay.

• The discussion section could be strenghten up with other studies carried out globally in other aging countries if possible.

Minor comments:

• General

o The language should be adjusted to use "low- and middle-income countries" instead of "developed" or "developing countries" for accuracy, impartiality, and to avoid perpetuating stereotypes and biases associated with the traditional terms.

• Abstract:

o Data Source Information: The abstract mentions using a locally designed SCI database, but it doesn't describe this database in detail. Readers might be interested in understanding the data collection methods, and its reliability.

• Introduction:

o Limited Context: While the introduction provides background information about TSCI and its historical trends, it lacks a comprehensive literature review. A thorough review of existing literature would help provide a better context for the study and demonstrate how the current study contributes to the field.

o The introduction goes from epidemiology to Japan and back to epidemiological data. Restructuring the introduction to commence with the definition, followed by an exploration of epidemiology and its associated costs, then delving into the relevance of the study within the local context of Japan (single center), addressing the knowledge gap, and finally elucidating the study's objectives and its broader significance, may enhance the overall introduction.

o The introduction can be edited to include a small section on some of the debilitating complications associated with TSCI in order to show the great effect such injuries have on the QOL of patients

• Materials and Methods:

o In order to enhance the methodology section and ensure comprehensive and transparent reporting, incorporating the STROBE guidelines to restructure the content is advisable. https://www.strobe-statement.org/checklists/

o Incomplete Data Description: The section provides details about data collection but lacks information about sample size justification, data quality control measures, and ethical considerations. These omissions affect the transparency and replicability of the study.

o It says: “May 2011 to December 2011, patient data could not be entered into the database because of hospital renovations; therefore, the number of registered patients decreased considerably in 2005 and 2011” but it is not clear how a renovation in May 2011 would affect the data in 2015. This is maybe a typo.

o Bias: The section should include a discussion of potential limitations of the study. For instance, the lack of complete data for certain years and the exclusion of pediatric cases are mentioned in the conclusion but should be included here for clarity.

o Statistical Analysis: trends should not be analyzed with chi-squared test. More elaborated analysis including regressions (going from linear regressions to time-series analysis) would help to really understand the differences in the periods and the rates of occurrence of the outcome variables, which should be more clearly specified.

o The chi square test ideally indicates significance of association and not the strength of association as mentioned in the statistical analysis section in the methods.

• Results:

o Considering the feedback provided in the methods section, it is evident that additional effort is needed, particularly in the statistical analysis portion of the results.

o Please harmonize the legends of Figure 1. There seems to be an error whereby E is captioned twice.

• Discussion:

o Incomplete Explanation of Findings: The discussion section provides some interpretation of the data but lacks a comprehensive analysis of the trends observed in the study. A more detailed discussion of how the changing demographics of TSCI patients might impact healthcare resources and strategies is needed.

o Minor grammatical mistakes in conclusion as highlighted.

o Please add the fact that this is a single center study and that limits generalizability. A similar issue was mentioned in the introduction when arguing that the incidence of TSCI varies from region to region even within the same country.

o Limited Discussion of Alcohol Use: Is this study really addressing this issue? While the study mentions the impact of alcohol on TSCI, there is a lack of discussion about potential interventions or policy recommendations related to alcohol use as a preventive measure.

o Future Research: The discussion section should conclude with suggestions for future research, building on the findings and limitations of the current study.

• Conclusion:

o Incomplete Summary of Findings: The conclusion should provide a concise summary of the key findings and their implications for the field of TSCI research and healthcare policy. This would help readers understand the study's significance more clearly.

o Please complete your conclusion. Line 4 has an incomplete sentence.

In summary, this scientific article provides valuable data on the epidemiology and demographics of TSCI in Japan, but it contains several flaws related to clarity, completeness, and context. Addressing these issues would enhance the quality and impact of the research.

6. PLOS authors have the option to publish the peer review history of their article (what does this mean?). If published, this will include your full peer review and any attached files.

Reviewer #1: No

Reviewer #2: No

---

## [Author Response · Author response to Decision Letter 0]

16 Dec 2023

In a separate document, we have addressed each reviewer’s comments in a point-by-point manner. We would appreciate it if you could review the contents of the attached "Response to Reviewers" file. The reviewers’ comments are highlighted in bold, and our responses are provided below. Changes made in the revised manuscript are indicated in italics and underlined. We believe that these revisions have significantly enhanced the overall quality of the manuscript. Should further revisions be necessary, I am more than willing to make the required adjustments.

---

## [Decision Letter · Decision Letter 1]

29 Jan 2024

PONE-D-23-30026R1Changing trends in traumatic spinal cord injury in an aging society: epidemiology of 1152 cases over 15 years from a single center in JapanPLOS ONE

Dear Dr. Yokota,

Thank you for submitting your manuscript to PLOS ONE. After careful consideration, we feel that it has merit but does not fully meet PLOS ONE’s publication criteria as it currently stands. Therefore, we invite you to submit a revised version of the manuscript that addresses the points raised during the review process.

We look forward to receiving your revised manuscript.

Kind regards,

Yih-Kuen Jan, PhD

Academic Editor

PLOS ONE

Journal Requirements:

**Additional Editor Comments:**

Both reviewers see the contribution from this study to the literature. They recommend to further the discussion by comparing the data from this study with similiar studies from other countries. The editor agrees that these comparisioins would increase the impact of this study.

Reviewers' comments:

Reviewer's Responses to Questions

**Comments to the Author**

1. If the authors have adequately addressed your comments raised in a previous round of review and you feel that this manuscript is now acceptable for publication, you may indicate that here to bypass the “Comments to the Author” section, enter your conflict of interest statement in the “Confidential to Editor” section, and submit your "Accept" recommendation.

Reviewer #1: All comments have been addressed

Reviewer #2: All comments have been addressed

2. Is the manuscript technically sound, and do the data support the conclusions?

Reviewer #1: Yes

Reviewer #2: Yes

3. Has the statistical analysis been performed appropriately and rigorously? 

Reviewer #1: Yes

Reviewer #2: Yes

4. Have the authors made all data underlying the findings in their manuscript fully available?

Reviewer #1: Yes

Reviewer #2: Yes

5. Is the manuscript presented in an intelligible fashion and written in standard English?

Reviewer #1: Yes

Reviewer #2: Yes

6. Review Comments to the Author

Reviewer #1: (No Response)

Reviewer #2: Dear Author,

Thank you for the revisions made to your manuscript, "Changing trends in traumatic spinal cord injury in an aging society: epidemiology of 1152 cases over 15 years from a single center in Japan."

The improvements are noticeable and commendable. However, to enhance the publishability of your article, I recommend expanding its scope beyond Japan, a high-income country, to include perspectives from low- and middle-income countries (LMICs). This would enrich the global epidemiological discussion and make your findings more relevant internationally. Incorporating studies like the one found at PubMed (https://pubmed.ncbi.nlm.nih.gov/34035224/), which discusses similar issues in South America and other LMICs, could provide valuable insights into how different socioeconomic contexts affect traumatic spinal cord injuries.

This addition would significantly broaden the appeal and applicability of your study.

7. PLOS authors have the option to publish the peer review history of their article (what does this mean?). If published, this will include your full peer review and any attached files.

Reviewer #1: No

Reviewer #2: No

---

## [Author Response · Author response to Decision Letter 1]

30 Jan 2024

In a separate document, we have addressed each reviewer’s comments in a point-by-point manner. We would appreciate it if you could review the contents of the attached "Response to Reviewers" file. The reviewers’ comments are highlighted in bold, and our responses are provided below. Changes made in the revised manuscript are indicated in italics and underlined. We believe that these revisions have significantly enhanced the overall quality of the manuscript. Should further revisions be necessary, I am more than willing to make the required adjustments.

---

## [Decision Letter · Decision Letter 2]

31 Jan 2024

Changing trends in traumatic spinal cord injury in an aging society: epidemiology of 1152 cases over 15 years from a single center in Japan

PONE-D-23-30026R2

Dear Dr. Yokota,

We’re pleased to inform you that your manuscript has been judged scientifically suitable for publication and will be formally accepted for publication once it meets all outstanding technical requirements.

Kind regards,

Yih-Kuen Jan, PhD

Academic Editor

PLOS ONE

Additional Editor Comments (optional):

Reviewers' comments:

Reviewer's Responses to Questions

**Comments to the Author**

1. If the authors have adequately addressed your comments raised in a previous round of review and you feel that this manuscript is now acceptable for publication, you may indicate that here to bypass the “Comments to the Author” section, enter your conflict of interest statement in the “Confidential to Editor” section, and submit your "Accept" recommendation.

Reviewer #2: All comments have been addressed

2. Is the manuscript technically sound, and do the data support the conclusions?

Reviewer #2: Yes

3. Has the statistical analysis been performed appropriately and rigorously? 

Reviewer #2: Yes

4. Have the authors made all data underlying the findings in their manuscript fully available?

Reviewer #2: Yes

5. Is the manuscript presented in an intelligible fashion and written in standard English?

Reviewer #2: Yes

6. Review Comments to the Author

Reviewer #2: I greatly appreciate your diligence in addressing the comments presented. Your dedication and effort in this endeavor are commendable, and I extend my sincere congratulations on the remarkable quality of your work.

7. PLOS authors have the option to publish the peer review history of their article (what does this mean?). If published, this will include your full peer review and any attached files.

Reviewer #2: No

---

## [Editor Report · Acceptance letter]

30 Apr 2024

PONE-D-23-30026R2 

PLOS ONE

Dear Dr. Yokota, 

I'm pleased to inform you that your manuscript has been deemed suitable for publication in PLOS ONE. Congratulations! Your manuscript is now being handed over to our production team.

Kind regards, 

on behalf of

Dr. Yih-Kuen Jan 

Academic Editor

PLOS ONE